# Individualized Pooled CRISPR/Cas9 Screenings Identify CDK2 as a Druggable Vulnerability in a Canine Mammary Carcinoma Patient

**DOI:** 10.3390/vetsci12020183

**Published:** 2025-02-18

**Authors:** Marine Inglebert, Martina Dettwiler, Chang He, Enni Markkanen, Lennart Opitz, Arunasalam Naguleswaran, Sven Rottenberg

**Affiliations:** 1Institute of Animal Pathology, Vetsuisse Faculty, University of Bern, 3012 Bern, Switzerland; inglebert.marine@gmail.com (M.I.); info@vetscope.com (M.D.); chang.he@unibe.ch (C.H.); arunasalam.naguleswaran@unibe.ch (A.N.); 2Graduate School for Cellular and Biomedical Sciences, University of Bern, 3012 Bern, Switzerland; 3Vetscope Pathologie Dettwiler, Lörracherstrasse 50, 4125 Riehen, Switzerland; 4Institute of Veterinary Pharmacology and Toxicology, Vetsuisse Faculty, University of Zürich, 8056 Zürich, Switzerland; enni.markkanen@vetpharm.uzh.ch; 5Functional Genomics Center Zurich, University of Zürich and ETH, 8092 Zürich, Switzerland; lopitz@fgcz.ethz.ch; 6Bern Center for Precision Medicine, University of Bern, 3012 Bern, Switzerland; 7Cancer Therapy Resistance Cluster, Department for BioMedical Research, University of Bern, 3012 Bern, Switzerland

**Keywords:** gene-editing tools, mammary tumor, patient-derived organoid (PDO), epigenome, druggable gene, precision medicine

## Abstract

This study explores the potential of canine patient-derived organoids (PDOs) in identifying mammary tumor-specific weaknesses by comparing tumor cells with matching normal mammary PDOs from the same patient. For this purpose, two customized CRISPR/Cas9 screens were performed to identify genes that are essential for tumor cells to survive but dispensable for normal mammary epithelial cells. CDK2 was subsequently identified as an important vulnerability in PDOs derived from a canine mammary tumor patient, which can be targeted by an existing inhibitor, PF3600. Although we only used tumors and normal PDOs from a single patient, our approach exemplifies how to discover novel, more individualized anti-cancer treatments for dogs.

## 1. Introduction

Canine mammary tumors (CMTs) rank second among the most frequent tumors in predominantly older, female dogs [1,2,3]. About half of the CMTs are malignant and display highly heterogenous histological subtypes [4]. They mainly arise from epithelial cells, but some also involve myoepithelial cells (complex tumors) or mesenchymal cells (mixed tumors). Pure mesenchymal or myoepithelial tumors rarely occur [5]. Frequently, multiple CMTs of different histological subtypes can occur in various mammary complexes of the same patient [6]. Due to shared key clinicopathologic features, dogs are considered to be a potential spontaneous cancer model for human breast cancer (HBC) in translational oncology [7,8]. Furthermore, mutations in the PI3K-Akt pathway are a frequent finding in both species. However, one of the main challenges in HBC research remains in the identification of biomarkers that can efficiently predict patients’ therapy responses [9]. Additionally, there are species-specific molecular mechanisms, such as, for example, fewer *TP53* mutations in CMT compared to HBC and a lower tumor mutational burden [10,11,12,13]. Furthermore, while the finding may be specific to the cohort analyzed, whole genome sequencing (WGS) analysis performed on a cohort of 21 malignant CMTs suggested that characteristic HBC mutational signatures of HBC, such as those resulting from defects in APOBEC cytidine deaminases or homologous recombination repair deficiency [14], do not appear to play a significant role in CMT carcinogenesis [11]. Instead, age-related deamination of methyl cytosines seems to be the dominant genotype. While CMTs are lower in their tumor mutational burden, their phenotypic heterogeneity suggests epigenetic or post-translational modifications to play a role, which could potentially explain the co-evolution of complex carcinomas with myoepithelial involvement [15].

In veterinary medicine, surgery remains the treatment of choice for CMTs, and the determination of biomarkers, such as hormone receptors, is not routinely performed in the clinic. Currently, those biomarkers are mainly used as prognostic indicators for research purposes [16,17], and no study links them to therapy response. This is mostly because chemotherapy often does not provide a clear advantage for dogs suffering from CMTs [5], and clinical studies are limited. For patients suffering from metastatic diseases, treatment options are often unsuccessful. Therefore, more efforts are needed to develop targeted therapies in canine patients. This requires solid preclinical studies to increase our understanding of CMT carcinogenesis and to identify therapeutic vulnerabilities.

For this purpose, we have previously generated models for patient-derived organoids (PDOs) from CMTs that accurately recapitulate the morphology, hormone receptor status, and genetic characteristics of their tissues of origin while maintaining disease heterogeneity [11]. In particular, CMT organoids derived from the malignant and non-neoplastic mammary epithelial tissues from the same patient can be established and expanded long-term [11]. One helpful approach to understanding tumorigenesis is identifying and characterizing genes that are essential (“essentialome”) for tumor cells to survive but are dispensable for normal cells. Those genes that are essential in tumor cells but not in normal tissues may be useful therapeutic targets with high tumor cell-specific effectiveness and minimal side effects (the concept of synthetic lethality) [18]. We previously demonstrated that CMT organoids can be used as an ex vivo tool for studying mutation mechanisms with CRISPR/Cas9 [11]. CRISPR/Cas9 libraries of many different individual perturbations, i.e., each single-guide RNA (sgRNA), offer high throughput screening possibilities to understand specific tumor behaviors [19,20]. Those libraries can be genome-wide or designed to study a selection of genes of interest or one specific pathway [21]. They can be used to develop novel therapeutic targets [22], but to our knowledge no CRISPR/Cas9 dropout screen compared a cancer cell line to its matched non-neoplastic tissue derived from the same patient. Using organoids, a genome-wide CRISPR/Cas9 screen in induced pluripotent stem cell-derived kidney organoids has been successfully performed [23]. For PDOs, however, scaling up is challenging, and the number of cells to reach the appropriate library coverage (i.e., the number of cells having one single sgRNA) is difficult to obtain [24]. In this context, the use of targeted libraries simplifies experimental setups. This technique has been applied to patient-derived colon organoids, where a custom pan-cancer tumor suppressor gene CRISPR library has been generated to identify tumor drivers [25].

In this study, we functionally identified the druggable vulnerabilities of CMTs using two independent pooled CRISPR/Cas9 dropout screening approaches for PDOs from both the carcinoma and non-neoplastic mammary tissues from an individual canine patient. Using this approach, we support the basic concept of unraveling specific predictive markers and guide treatment options for individual patients based on differential functional genetic screening using matched PDOs from neoplastic versus non-neoplastic tissues.

## 2. Materials and Methods

### 2.1. Sample Collection and Tissue Processing

We previously generated a living biobank of CMTs and non-neoplastic mammary tissue from client-owned donor dogs [11]. Briefly, surplus mammary tissue from dogs undergoing standard mastectomy due to spontaneous CMTs were obtained in collaboration with small animal veterinary clinics throughout Switzerland between 2018 and 2020. All procedures followed standard care veterinary guidelines and relevant Swiss regulations and ethical standards. Informed consent was obtained for each sample in accordance with the ARRIVE guidelines (https://arriveguidelines.org/ accessed on 22 September 2022) when applicable. Ethical approval was granted by the Cantonal Veterinary Office. Additional EDTA blood for DNA isolation was collected with approval by the “Cantonal Committee for Animal Experiments” (Canton of Bern; permit 71/19).

Half of the CMTs were formalin-fixed paraffin-embedded (FFPE) while the remaining tissues were minced and cryopreserved using a freezing medium (10% DMSO, 45% Dulbecco’s Modified Eagle Medium, 45% Fetal Calf Serum (Thermo Fisher Scientific, Waltham, MA, USA)). Randomly selected pieces were snap-frozen and stored at −80 °C for RNA isolation.

### 2.2. Histology

Hematoxylin and eosin (H&E)-stained FFPE sections of primary tumors were analyzed by a board-certified veterinary pathologist (M.D.). Primary tumors were classified and graded according to the commonly accepted Davis–Thompson classification and grading system [15,17].

### 2.3. Maintenance of CMT Organoid Cultures

Using our previously established biobank of canine mammary tumors, we selected two organoid lines derived from both normal and cancerous tissue with known long-term culture stability for this study. Both PDOs were derived from the same treatment-naive 9-year-old intact female Cocker Spaniel, which underwent radical unilateral mastectomy due to multiple benign and malignant mammary neoplasms. ORG-63-C was derived from a nodule histopathologically diagnosed as a complex carcinoma while adjacent grossly non-neoplastic glandular tissue was used to derive ORG-63-N [11].

Frozen organoids were thawed, rinsed, and the pellet was resuspended in cold CMT organoid medium [11] and mixed at a 1:1 ratio with Cultrex^®^ PathClear Reduced Growth Factors Basement Membrane Extract (BME) Type 2 (Amsbio, Abingdon, UK). The BME-cell suspension was seeded as 30 µL drops on prewarmed 24-wells suspension culture plates (Greiner Bio-One, Kremsmünster, Austria) and cultured following standard protocols [11,26] to expand the organoids.

### 2.4. Lentivirus Production, Lentiviral Transduction

Lentivirus was produced using standard protocols [27] by delivering pLentiCRISPRv2 vectors loaded with the custom sub-libraries and lentiviral packaging plasmids. Transduction of the organoids was performed according to previous protocols [28].

### 2.5. Pooled CRISPR/Cas9 Screening

Both lentiviral-based CRISPR/Cas9 sub-libraries were designed using the canine CanFam3.1 assembly. Sub-library CP1736 contains 6004 sgRNAs, targeting 834 “druggable” genes with known inhibitors (six sgRNAs per gene) along with 500 non-targeting and intergenic controls each (see Appendix A). Sub-library CP1737 contains 8614 sgRNAs, targeting 1269 epigenetic regulatory genes (six sgRNAs per gene) with 500 non-targeting and intergenic controls (see Appendix A). For each screen, sixteen million ORG-63-C and ORG-63-N were collected and transduced with lentivirus at a multiplicity of infection (number of viral particles per cell) of four. The medium was replaced with a puromycin-containing medium (3.5 µg/mL, GIBCO) 24 h later, and selection was performed for eleven days. Twelve days after transduction (day 0 = D0), organoids were trypsinized, a pool of cells isolated for D0 (at least 3 million), and we seeded the rest of the cells into two technical replicates (R1, R2) with a coverage of 500 cells per sgRNA (at least 3 million cells per replicate). We passaged the organoids every ten days and harvested them after ten doubling times, hence after forty days (D40). DNA was extracted from the D0 and D40 population and sent for sequencing. The sgRNAs that were differentially enriched or depleted after 40 days compared to the pDNA were identified with the MAGeCK software version 0.5 [29]. For PCR amplification, gDNA was divided into 100 μL reactions such that each well had at most 10 μg of gDNA.

For each 96-well plate, the PCR master mix was prepared by combining 150 μL of DNA Polymerase (Titanium Taq; Takara, Kusatsu, Japan), 1 mL of 10× buffer, 800 μL of dNTPs (Takara), 50 μL of P5 stagger primer mix (AATGATACGGCGACCACCGAGATCTACACTCTTTCCCTACACGACGCTCTTCCGATCT[s]TTGTGGAAAGGACGAAAC*A*C*C*G, where [s] represents the barcode region, 100 μM), and the remaining volume was adjusted to 4 mL with water. Each well contained 50 μL of gDNA plus water, 40 μL of PCR master mix, and 10 μL of a uniquely barcoded P7 primer (CAAGCAGAAGACGGCATACGAGAT[s]GTGACTGGAGTTCAGACGTGTGCTCTTCCGATCTCCAATTCCCACTCCTTTCAAG*A*C*C*T, where [s] is the barcode region, 5 μM). The PCR cycling conditions were as follows: (1) 95 °C for 1 min; (2) 94 °C for 30 s; (3) 52.5 °C for 30 s; (4) 72 °C for 30 s; (5) go to (2), 27 cycles; (6) 72 °C for 10 min. PCR products were purified using Agencourt AMPure XP SPRI beads, following the manufacturer’s protocol (Beckman Coulter, Beverly, MA, USA A63880). Samples were sequenced on a HiSeq2500 High Output (Illumina, Ipswich, MA, USA)) with a 5% PhiX spike-in. Enrichment analysis was performed using MAGeCK (Model-based Analysis of Genome-wide CRISPR/Cas9 Knockout) [29].

### 2.6. Proliferation Assay

Organoids were dissociated into single cells, and 35,000 cells were seeded per well in 10 µL BME/CMT medium drops on a 24-wells plate. Every day, proliferation was assessed using the resazurin-based Cell Titer Blue following the manufacturer’s instructions (Promega, Madison, WI, USA). In brief, 25 µL of the reagent was added to the culture medium and incubated for 4 h at 37 °C. 200 µL of the medium was then pipetted to a 96-wells plate, and fluorescence intensity at 560Ex/590Em nm was determined with an EnSpire Multimode Plate Reader (PerkinElmer, Waltham, MA, USA). The mean and standard deviation of three different wells was calculated. Two technical replicates were performed.

### 2.7. Drug Testing and Cell Viability Assays

Per well in a 24-well plate, a droplet of 10 µL single-cell-suspension containing 35,000 dissociated organoid cells with BME and medium was seeded. After polymerization of the BME/medium droplet, medium infused with various concentrations of PF-06873600 (AdipoGen Life Sciences, Füllinsdorf, Switzerland), doxorubicin (Teva Pharma AG, Basel, Switzerland), reversine R3904 (Sigma Aldrich, St. Louis, MO, USA), and luminespib NVP-AUY922 (Selleckchem, Houston, TX, USA) was added to the well without disrupting the droplet. The growth medium containing doxorubicin was replaced with a drug-free medium after 24 h, while the medium was refreshed after 72 h of incubation for the remaining treatments. After six days of incubation, cell viability was assessed using a resazurin-based dye (Cell Titer Blue, Promega) in at least two biological replicates per treatment. Untreated controls were used for normalization, and IC50 values were determined by fitting cell growth data to a four-parameter logistic sigmoidal curve.

### 2.8. RNA Isolation and Sequencing of the Primary Tumors and Non-Neoplastic Mammary Tissues

Sample and library preparation: The list of tissues sequenced can be found in Appendix A. In total, RNA from 34 carcinomas and 17 non-neoplastic mammary tissues were isolated from snap-frozen samples stored at −80 °C using the RNeasy mini kit (Qiagen, Hilden, Germany) following the manufacturer’s instructions. RNA sequencing was performed at the Novogene company. RNA degradation and contamination were monitored on 1% agarose gels. RNA purity was checked using the NanoPhotometer^®^ spectrophotometer (IMPLEN, Westlake Village, CA, USA). RNA integrity and quantitation were assessed using the RNA Nano 6000 Assay Kit of the Bioanalyzer 2100 system (Agilent Technologies, Santa Clara, CA, USA). A total amount of 1 µg RNA per sample was used as input material for the RNA sample preparations. Sequencing libraries were generated using NEBNext^®^ Ultra™ RNA Library Prep Kit for Illumina^®^ (NEB, Ipswich, MA, USA) following the manufacturer’s recommendations, and index codes were added to attribute sequences to each sample. Briefly, mRNA was purified from total RNA using poly-T oligo-attached magnetic beads. Fragmentation was carried out using divalent cations under an elevated temperature in NEBNext First Strand Synthesis Reaction Buffer (5X) or by using sonication with Diagenode Bioruptor Pico for breaking RNA strands. First-strand cDNA was synthesized using a random hexamer primer and M-MuLV Reverse Transcriptase (RNase H-). Second strand cDNA synthesis was subsequently performed using DNA Polymerase I and RNase H. Remaining overhangs were converted into blunt ends via exonuclease/polymerase activities. After adenylation of 3′ ends of DNA fragments, NEBNext Adaptor with hairpin loop structure was ligated to prepare for hybridization. To select cDNA fragments of preferentially 150–200 bp in length, the library fragments were purified with the AMPure XP system (Beckman Coulter, Beverly, MA, USA). Then, 3 µL USER Enzyme (NEB, USA) was used with size-selected, adaptor-ligated cDNA at 37 °C for 15 min, followed by 5 min at 95 °C before PCR. Then, PCR was performed with Phusion High-Fidelity DNA polymerase, Universal PCR primers, and Index (X) Primer. At last, PCR products were purified (AMPure XP system), and library quality was assessed on the Agilent Bioanalyzer 2100 system. Then, the clustering of the index-coded samples was performed on a cBot Cluster Generation System using PE Cluster Kit cBot-HS (Illumina) according to the manufacturer’s instructions. After cluster generation, the library preparations were sequenced on an Illumina platform, and paired-end reads were generated.

Data analysis: Raw data (raw reads) of FASTQ format were firstly processed using fastp (Version 0.20) [30]. For the downstream analysis, adapter content, poly-N sequences, and low-quality reads were removed to generate clean reads, and Q20, Q30, and GC content were calculated. Reference genome and gene annotation files were retrieved from genome browsers, such as NCBI, UCSC, or Ensembl. The paired-end clean reads were then aligned to the reference genome using HISAT2 software v2.0.5. The mapped reads of each sample were assembled by StringTie (vl.3.3b) [31]. The featureCounts vl.5.0-p3 program was used to count the reads numbers mapped to each gene. FPKM (expected number of Fragments Per Kilo base of transcript sequence per Millions base pairs sequenced of each gene) was calculated based on the length of the gene and reads count mapped to this gene. Before differential gene expression analysis, for each sequenced library, the read counts were adjusted by Trimmed Mean of M-values through one scaling normalized factor. Then, differential expression analysis was performed using the edgeR R package (3.22.5). The *p*-values were adjusted using the Benjamini–Hochberg method. The corrected *p*-value of 0.05 and absolute fold change of 2 were set as the threshold for significantly differential expression.

Enrichment analysis: Gene Ontology (GO) enrichment analysis of differentially expressed genes was implemented by the clusterProfiler R package, correcting gene length bias. GO terms with corrected *p*-value less than 0.05 were considered significantly enriched by differential expressed genes. Kyoto Encyclopedia of Genes and Genomes (KEGG) Pathway enrichment analysis was performed by the clusterProfiler R package to test the statistical enrichment of differential expression genes.

### 2.9. RNA Sequencing of the Organoids (Paired Carcinoma and Non-Neoplastic Mammary Tissue)

Sample and library preparation: Organoids were trypsinized, and RNA was extracted using the RNeasy mini kit (Qiagen) following the manufacturer’s instructions. The quality of the isolated RNA was determined with a Fragment Analyzer (Agilent, Santa Clara, CA, USA). Only the samples with a 260 nm/280 nm ratio between 1.8 and 2.1 and a 28S/18S ratio within 1.5–2 were further processed. The TruSeq Stranded mRNA (Illumina, Inc., San Diego, CA, USA) was used in the succeeding steps. Briefly, total RNA samples (100–1000 ng) were poly-A enriched and then reverse-transcribed into double-stranded cDNA. The cDNA samples were fragmented, end-repaired, and adenylated before ligation of TruSeq adapters containing unique dual indices for multiplexing. Fragments containing TruSeq adapters on both ends were selectively enriched with PCR. The quality and quantity of the enriched libraries were validated using the Fragment Analyzer (Agilent, Santa Clara, CA, USA). The product is a smear with an average fragment size of approximately 260 bp. The libraries were normalized to 10 nM in Tris-Cl 10 mM, pH8.5 with 0.1% Tween 20. The NovaSeq 6000 (Illumina, Inc, San Diego, CA, USA) was used for cluster generation and sequencing according to standard protocol. The sequencing mode was a single end with 100 bp.

Data analysis: The raw reads were first cleaned by removing adapter sequences, trimming low-quality ends, and filtering reads with low quality (phred quality < 20) using fastp (Version 0.20) [30]. Sequence pseudo alignment of the resulting high-quality reads to the reference genome (build CanFam3.1, gene models based on Ensembl release 104 downloaded on 1 June 2021) and quantification of gene-level expression was carried out using Kallisto (Version 0.46.1) [32].

When comparing RNA sequencing data from primary tissues and organoids, the raw reads were cleaned by removing adapter sequences, trimming low-quality ends, and filtering low-quality reads using tools available in the Galaxy platforms (usegalaxy.org and usegalaxy.org.au). Subsequently, reads were mapped to the canine genome (Canis_lupus_familiaris.ROS_Cfam_1.0.dna.toplevel) using the bowtie2 tool with default parameters. Mapped reads counts to protein-coding transcripts were extracted from the BAM file based on the gene annotations from the Canis_lupus_familiaris.ROS_Cfam_1.0.107.gff3 file using the featureCounts tool. To detect differentially expressed genes, we applied a count-based negative binomial model implemented in the software package DESeq2 (R version: 4.2.0, DESeq2 version: 1.36.0) [33]. Genes showing altered expression with an adjusted (Benjamini–Hochberg method) *p*-value < 0.05 were considered differentially expressed.

### 2.10. Statistical Analysis and Data Representation

Statistical analyses and data representation were mainly performed using Prism statistical software (v9.0; GraphPad Inc, San Diego, CA, USA). Statistical tests and *p*-values are indicated in the text or the figures’ legends. Parts of some figures were created with BioRender.com and Inkscape (RRID: SCR_014479). The heatmaps of Appendix A were designed with Morpheus (https://software.broadinstitute.org/morpheus accessed on 22 September 2022).

## 3. Results

### 3.1. Design and Lentiviral Transduction of Two Custom Canine CRISPR/Cas9 Libraries to Identify Therapeutic Vulnerabilities in CMTs

To assess which genes are essential in mammary tumor cells but not in adjacent normal mammary tissues, we used two organoid (ORG) lines from the same patient, called dog 63 (ORG-63-N, derived from non-neoplastic mammary tissue and ORG-63-C, derived from a mammary carcinoma). We transduced those two organoid lines with different customized canine CRISPR/Cas9 sub-libraries following standard guidelines for sgRNA optimization (Figure 1a) [34,35]. Since we did not observe many genetic alterations in CMTs that would explain tumorigenesis [11], we hypothesized that epigenetic alterations might play a crucial role in the formation of CMTs. Therefore, we designed the first custom library to target canine genes that may be essential for the epigenetic regulation of tumors but not normal mammary epithelial cells. This canine “epigenome” library (CP1737) contains 8614 sgRNAs, targeting 1269 genes (six sgRNAs/gene, see Appendix A), of which the human homologs have previously been linked to epigenetic processes [35,36,37]. We cloned this library into the lentiCRISPRv2 one vector system containing Cas9 (pXPR_023). Moreover, as many genes involved in epigenetic regulation do not have known inhibitors, and as our main aim was to find concrete new treatment options for CMTs, we designed a second custom library that targets “druggable” genes, i.e., with known inhibitors existing. The “druggable” library (CP1736) contains 6004 sgRNAs, targeting 834 canine genes (six sgRNAs/gene, see Appendix A), of which the human or mouse homologs have been described to be druggable by already existing compounds [35,38,39]. We also cloned this library into the lentiCRISPRv2 one vector system containing Cas9 (pXPR_023).

For both independent screening experiments, we aimed to express the library in CMT-derived organoids at a coverage of 500 cells per sgRNA. After twelve days of puromycin selection (necessary for the organoids to recover from the transduction and survive the following trypsinization), all read counts at D0 were higher than for the original plasmid DNA (pDNA) in both screens, indicating that the PCR reaction worked efficiently (see Appendix A). Importantly, we reached the 500x coverage of the libraries we aimed to obtain. After splitting the organoids into single cells at D0, we seeded two technical replicates (R1, R2) of the screen with a coverage of 500 cells per sgRNA. We passaged the organoids every ten days and harvested them after ten doubling times, i.e., after forty days (D40). Both cell lines show similar proliferation rates in vitro (see Appendix A). DNA was extracted from the D0 and D40 population and sent for sequencing. The sgRNAs that were differentially enriched or depleted after 40 days compared to the pDNA were identified with the MAGeCK software [29].

The quality of the two screens was further assessed by principal component analysis and calculating the Pearson correlation of the different samples to each other, based on the normalized read counts of all samples (see Appendix A). When we carried out a principal component analysis to compare the samples, we observed that the D0 samples cluster was very close to the pDNA. By contrast, the technical replicates at D40 (R1, R2) of ORG-63-N and ORG-63-C cluster to each other, indicating that our analysis is reproducible for both screens (see Appendix A). In addition, the technical replicates at D40 of ORG-63-N and ORG-63-C were highly correlated for both screens, strengthening the reproducibility of our analysis (see Appendix A). Hence, sub-library screens using about 6000–9000 sgRNAs are technically feasible in CMT-derived organoids.

### 3.2. Pooled CRISPR/Cas9 Screening Targeting the Canine Epigenome in PDOs to Better Understand CMT Biology and Tumorigenesis

The first screen aimed to understand which epigenetic alterations may be crucial for the growth of CMTs but dispensable for normal mammary epithelial cells derived from the same individual patient. When comparing sgRNA enrichment between D40 and pDNA, some sgRNAs were clearly enriched for both organoid lines (log fold change = LFC > 1 and *p*-value < 0.05), matching genes known to bring a survival benefit for the cells when the gene is knocked-out, like SETD2 (coding for a histone methyltransferase containing a transcriptional activation domain) and CDKN2A (Cyclin Dependent Kinase Inhibitor 2A) (labeled in red in Figure 1c and Appendix A). Regarding “drop-off”, sgRNA targeting genes involved in essential processes, such as SRSF1 (splicing factor) or POLR2 (RNA polymerase II), were depleted in both organoid lines (core-specific essential genes, labeled in green in Figure 1c, left part of Figure 1e and Appendix A). Based on previous essentialome screens using human cell lines, these canine homologs were expected to come up as significant hits [40], further supporting the good quality of the screen.

In this “epigenome” screen, we identified 133 genes to be significantly depleted (LFC < −1 and *p*-value < 0.05) in the carcinoma organoid line (ORG-63-C) and 61 genes to be significantly depleted in the non-neoplastic mammary organoid line (ORG-63-N) (see Appendix A). We were particularly interested in the genes essential for ORG-63-C but not essential for ORG-63-N (LFC > −1 or *p*-value > 0.05) (context-specific essential genes, in bold in Appendix A). Targeting those genes may lead to carcinoma cell death while preventing damage to the non-neoplastic cells. This comparison yielded 45 candidates listed in Table 1 (see also the genes labeled in orange in Figure 1c, right part of Figure 1e, and Appendix A).
Figure 1Two independent pooled CRISPR/Cas9 screening approaches in paired CMT organoids identified therapeutic vulnerabilities of canine mammary carcinoma.
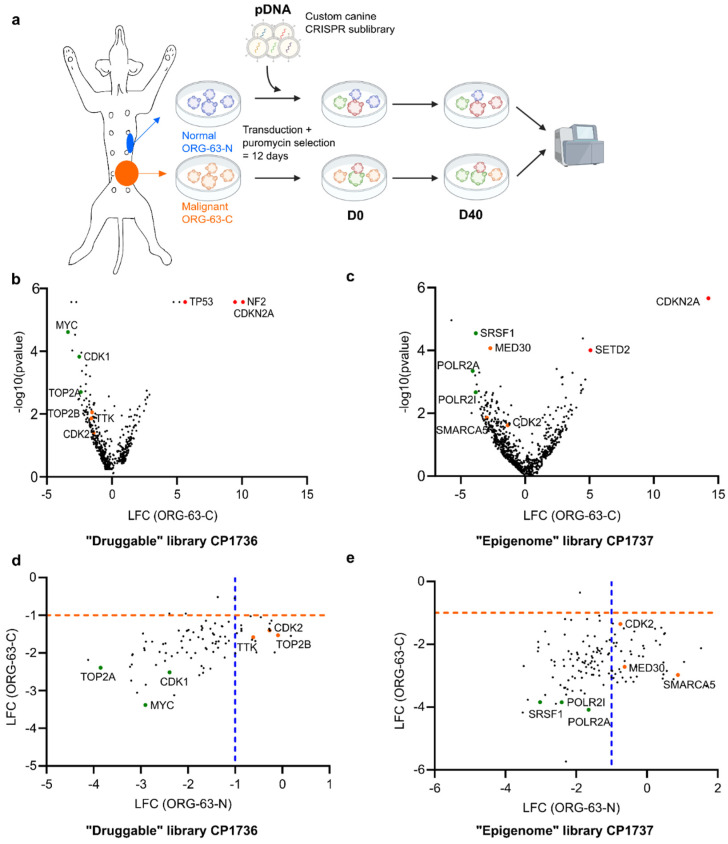

Outline of the screens performed with a custom canine CRISPR/Cas9 libraryVolcano plot representing depleted (LFC < 0) and enriched (LFC > 0) genes for ORG-63-C forty days (D40) after D0 for the druggable screen. LFC and *p*-values were calculated from two technical replicates with MAGeCK analysis. Each dot represents one gene for which at least four sgRNAs (out of six) were enrolled in the analysis. Selected hits are color-coded.Volcano plot representing depleted (LFC < 0) and enriched (LFC > 0) genes for ORG-63-C forty days (D40) after D0 for the epigenome screen. LFC and *p*-values were calculated from two technical replicates with MAGeCK analysis. Each dot represents one gene for which at least four sgRNAs (out of six) were enrolled in the analysis. Selected hits are color-coded.Scatter plot of the log fold change (LFC) of ORG-63-C and ORG-63-N for the druggable screen. Each dot represents one gene for which at least four sgRNAs (out of six) were enrolled in the analysis, and the *p*-value < 0.05 for at least one of the two ORG lines. Selected hits are color-coded.Scatter plot of the log fold change (LFC) of ORG-63-C and ORG-63-N for the epigenome screen. Each dot represents one gene for which at least four sgRNAs (out of six) were enrolled in the analysis, and the *p*-value < 0.05 for at least one of the two ORG lines. Selected hits are color-coded.
vetsci-12-00183-t001_Table 1Table 1Hits essential in ORG-63-C but dispensable in ORG-63-N (epigenome screen).GeneGene Description*p*-Value_Nlfc_N*p*-Value_Clfc_CSPENMsx2-interacting protein0.91411.5310.02454−2.126VPS72Vacuolar protein sorting-associated protein 72 homolog0.89790.46750.01602−2.577PLOD2Procollagen-lysine,2-oxoglutarate 5-dioxygenase 20.85991.68720.01308−3.333NCOR1Nuclear receptor co-repressor 10.76820.257540.021244−1.889SRSF2Serine/arginine-rich splicing factor 20.68090.468040.015419−2.955HDAC1Histone deacetylase 10.66390.509990.045327−1.647ACTL6AActin-like protein 6A0.603470.488820.009406−1.670TBL1XF-box-like/WD repeat-containing protein TBL1X0.60168−0.44380.046566−2.318BAZ1ABromodomain adjacent to zinc finger domain protein 1A0.59139−0.22590.014092−2.371NOC2LNucleolar complex protein 2 homolog0.58843−0.46930.018451−2.221TAF8Transcription initiation factor TFIID subunit 80.58046−0.35210.018177−1.594TFDP1Transcription factor Dp-10.469310.37270.033807−2.012CBX5Chromobox protein homolog 50.451740.405840.039584−1.473SMARCA5SWI/SNF-related matrix-associated actin-dependent0.444090.878410.013878−2.98SRRM1Serine/arginine repetitive matrix protein 10.43516−0.43180.019006−1.859COPS5Information COP9 signalosome complex subunit 50.432850.754870.015724−3.111NAP1L1Nucleosome assembly protein 1-like 10.40623−0.30090.024866−2.719HASPINSerine/threonine-protein kinase haspin0.40398−0.59460.007778−3.185RCC1Regulator of chromosome condensation0.394690.235470.01764−1.503OIP5Protein Mis18-beta0.38418−0.72620.018399−2.218SRSF5Splicing factor, arginine/serine-rich 4/5/60.34631−0.85430.04839−1.951MIS18BP1Mis18-binding protein 10.34095−1.00510.019045−2.84RBPMSRNA binding protein, mRNA processing factor0.33435−0.93470.047674−1.959SETD1AHistone-lysine N-methyltransferase SETD1A0.32603−0.64910.020014−1.490CDK2Cyclin-dependent kinase 20.31995−0.74950.024024−1.354USP7Ubiquitin carboxyl-terminal hydrolase 70.31791−1.05290.027406−1.860SUZ12SUZ12 Polycomb repressive complex 2 subunit0.31684−0.45370.019363−3.057DHX38Pre-mRNA-splicing factor ATP-dependent RNA helicase PRP160.31248−0.62690.025534−2.534KDM8JmjC-domain-containing protein 50.30885−0.88320.007473−2.061OTUB1Ubiquitin thioesterase0.30217−0.90420.017099−1.689GTF3C4General transcription factor 3C polypeptide 40.29667−0.10290.048652−2.046CENPAHistone H3-like centromeric protein A0.2798−0.89150.008441−2.949ZZZ3ZZ-type zinc finger-containing protein 30.26212−0.83310.017243−3.213CHEK1Serine/threonine-protein kinase Chk10.257331.05020.02631−2.563DDX17Probable ATP-dependent RNA helicase DDX170.248140.564020.031922−2.85SNAPC4snRNA-activating protein complex subunit 40.24499−0.97960.021048−2.847MED30Mediator Complex Subunit 300.17747−0.62678.51 × 10^−5^−2.719PCNAProliferating cell nuclear antigen0.14632−0.68460.012683−3.062ECDProtein ecdysoneless homolog0.12707−0.28610.026677−3.026RSF1Remodeling and spacing factor 10.11897−0.74640.032301−2.881UBE2AUbiquitin-conjugating enzyme E2 A0.11589−0.96190.010972−1.337GTF2BTranscription initiation factor IIB0.1149−0.77960.023854−2.791SMC2Structural maintenance of chromosomes protein 20.10679−0.38090.006639−3.191RUVBL1RuvB-like 10.0963280.446030.0039294−3.093AURKBAurora kinase B0.068751−0.94290.0055614−3.071

A string analysis network [41] of those candidates revealed that they are involved in many different biological processes, such as chromatin or chromosome organization, histone modification, transcription regulation, or organelle organization (118 Gene Ontology (GO) Processes involved in total, see Appendix A). The Kyoto Encyclopedia of Genes and Genomes (KEGG) pathways involved were cell cycle and viral carcinogenesis [42].

Overall, this epigenome screen identified numerous candidates involved in critical biological processes. Validating those hits may reveal interesting pathways worth exploring to understand CMT carcinogenesis better.

### 3.3. Pooled CRISPR/Cas9 Screening in PDOs to Identify Potential Drug Targets for CMTs

As many of those hits identified in the epigenome screen do not have known inhibitors, and our main aim was to find concrete new treatment options for CMTs, we moved forward to our second screening approach. This second screen aimed to identify actual therapeutic vulnerabilities of CMTs for which inhibitors exist to develop new therapeutic options for CMTs. For this purpose, we investigated which “druggable” genes are essential in PDOs derived from mammary carcinoma tissue but not in PDOs derived from normal mammary tissue.

When comparing sgRNA enrichment between D40 and pDNA, some sgRNAs were clearly enriched for both organoid lines (LFC > 1 and *p*-value < 0.05), matching genes known to bring a survival benefit for the cells when the gene is knocked-out, like TP53, NF2 (Neurofibromin-2), and CDKN2A (labeled in red in Figure 1b and Appendix A). Moreover, genes involved in essential processes, such as the proto-oncogene MYC or DNA Topoisomerase II Alpha TOP2A, were depleted in both organoid lines (core-specific essential genes, labeled in green in Figure 1b, left part of Figure 1d and Appendix A). Those genes, for which the behavior was expected based on published human essentialome screens [40], represented an additional quality control of the screen.

For the “druggable” screen, we identified 87 genes significantly depleted (LFC < −1 and *p*-value < 0.05) in the carcinoma organoid line (ORG-63-C) and 65 genes significantly depleted in the non-neoplastic mammary organoid line (ORG-63-N) (see Appendix A). We were particularly interested in the genes essential for ORG-63-C but not essential for ORG-63-N (LFC > −1 or *p*-value > 0.05) (context-specific essential genes, in bold in Appendix A). Indeed, targeting those candidates may result in neoplastic cell death without damaging the healthy cells. The 17 candidates from this analysis are listed in Table 2 (see the genes labeled in orange in Figure 1b, right part of Figure 1d, and Appendix A).

A string analysis network [41] of those candidates revealed that they are involved in many different biological processes, such as mitotic cell cycle and regulation, DNA repair, and DNA replication (40 Gene Ontology Processes involved in total, see Appendix A). The KEGG pathways involved were the Fanconi anemia pathway, cell cycle, PI3K-Akt signaling pathway, measles, p53 signaling pathway, human T-cell leukemia virus 1 infection, prostate cancer, small cell lung cancer, pathways in cancer, base excision repair, DNA replication, NOD-like receptor signaling pathway, viral carcinogenesis, nucleotide excision repair, Epstein–Barr virus infection, and shigellosis [42].

In summary, this “druggable” screen identified several candidates involved in different key biological processes, which may be attractive targets for CMT treatment. Both screens highlighted the increased essentiality of some genes in the neoplastic organoid line compared to the non-neoplastic organoid line. Notably, CDK2 appeared as a candidate overlapping between the two screens, which strengthened its biological relevance and made it a particularly compelling target for further investigation. We then wondered if this increased essentiality of some genes in the neoplastic organoid line compared to the non-neoplastic organoid line might also be represented in a difference in gene expression levels between primary tumor tissues and non-neoplastic mammary tissues.

### 3.4. Candidates from the CRISPR/Cas9 Screens Are Expressed in Primary Tumor Tissues and PDOs but Very Rarely Differentially Expressed Compared to the Non-Neoplastic Mammary Tissues

To obtain more insights into CMT biology and investigate how our different hits were expressed in normal and carcinoma mammary tissues, we first performed RNA sequencing of 34 carcinomas and 17 non-neoplastic mammary tissues (see Appendix A). Among these samples, we had 15 pairs (matched carcinoma/healthy mammary tissue) from 10 different patients (some dogs presented different tumors), including the pair the organoids for the screens were derived from (D63_3_CMT/D63_nor). The quality controls of the samples were satisfactory. Then, gene expression level determination (assessed by calculating the FPKM [expected number of fragments per kilobase of transcript sequence per millions base pairs sequenced], which takes the effects of both sequencing depth and gene length into consideration) and differential gene expression analysis could be performed. After assessing the gene expression level, we hierarchically clustered our samples depending on the gene expression level (see Appendix A). Together with this clustering, principal component analysis of all sequenced samples generated using FPKM (see Appendix A) revealed that a pool of non-neoplastic mammary tissues cluster very distinctly from the rest of the samples (D18_nor, D15_nor, D17_nor, D48_nor, D20_nor, D31_nor). However, the rest of the non-neoplastic mammary tissues cluster together with the carcinoma samples (D57_nor, D32_nor, D37_nor, D49_nor, D6_nor, D63_nor, D44_nor, D62_nor, D46_nor, D36_nor, D60_nor). Most likely, this is due to the gene expression derived from the non-neoplastic stromal cells that are also collected with the carcinoma samples. Indeed, differential gene expression between those two groups of normal mammary tissues showed that genes involved in immune response, leucocyte activation, and inflammatory response are upregulated in the normal tissues that cluster close to the carcinomas. As carcinomas are surrounded by inflammatory cells (inflammatory tumor microenvironment), this might explain why those two populations cluster together [43,44]. By contrast, the other pool of normal mammary tissues that clusters very distinctly from the rest of the samples is probably composed of non-neoplastic epithelial cells without an inflammatory component. Another explanation might be that some mammary tissues defined macroscopically as “normal” already had genetic aberrations similar to those in the CMTs. Indeed, carcinogenesis in CMTs is thought to follow a stepwise evolution from hyperplastic lesions to adenomas to carcinomas [6]. Therefore, sequencing adenoma samples and comparing their gene expression levels with this pool of “normal” samples that cluster close to the carcinomas might shed light on another possible reason for this clustering difference.

After analyzing the differential gene expression between all CMT samples compared with all normal mammary tissue samples, we found that 2831 genes are significantly (*p*-value < 0.05 and |LFC| > 1.0) upregulated (labeled in red in Figure 2a) and 3472 are significantly downregulated (labeled in green in Figure 2a). Among the hits that are upregulated, many genes involved in metabolic pathways can be found (see Appendix A), as well as genes coding for collagen or adhesion proteins, such as cadherins, claudins, or hepatic and glial cell adhesion molecules. Additionally, keratins and keratin-associated proteins are strongly downregulated in tumor tissues (labeled in blue in Figure 2a). We then investigated whether the candidate genes from the CRISPR screens that may be essential for tumor growth, but not for the growth of normal mammary epithelial cells, were expressed at higher levels in CMT tissues compared to normal tissues. Appendix A details the LFC and *p*-value for the candidates of the CRISPR screens in this RNA sequencing analysis of all CMTs compared to all normal tissues. Most of the candidates functionally essential for ORG-63-C, but not for ORG-63-N, are found to be expressed in the CMTs, showing they are not just induced by the in vitro culture conditions. However, they do not appear differentially expressed (*p*-value < 0.05 and |LFC| > 1.0) in the tumor tissues compared to normal tissues (see Appendix A). In fact, only two of our candidates from the epigenome screen (PLOD2 and CENPA) are upregulated when performing a differential gene expression analysis between the tissue ORG-63-C is derived from (D63_3_CMT) compared to the non-neoplastic tissue ORG-63-N is derived from (D63_nor) (labeled in orange in Figure 3b). When focusing on CDK2, the common candidate from both screens, we found that CDK2 was expressed in tumor tissues and normal mammary tissue samples. Moreover, when comparing all tumor tissues to all healthy mammary tissue samples, CDK2 was slightly downregulated (with an LFC of −0.63 and a *p*-value of 0.009, see Appendix A). However, this difference was not significant when focusing on the single pair D63_3_CMT/D63_nor.

We then set out to investigate the gene expression of our candidates in the organoid lines themselves. In these, stromal cells have been depleted due to the culture conditions, and one can compare the expression profiles of the neoplastic versus the non-neoplastic epithelium. For this purpose, we performed RNA sequencing of three matched pairs of organoids, including ORG-63-N and ORG-63-C. The other two were ORG-60-N/ORG-60-C (from dog D60) and ORG-17-N/ORG-17-C (from dog D17). The quality controls of the samples were satisfactory, and gene expression level and differential gene expression analysis could be performed. In addition, the principal component analysis generated using FPKM of all sequenced organoid samples (see Appendix A) revealed that dogs clustered together rather than according to their epithelial origins (carcinoma or non-neoplastic). This shows that the individual gene expression fingerprints of the epithelial cells are predominant over the gene expression of the tumor organoids versus the normal organoids.

Additionally, we compared the gene expression levels of primary tissues and organoids. Pearson’s correlation heat map (generated using normalized reads per million mapped reads) shows that the organoid samples cluster separately from all primary tissues (see Appendix A). Hence, although tumor-derived organoids keep features of the original tumor, as we have previously shown [11], the culture conditions still induce a gene expression signature that differs considerably from the original tissues. This raises the question of whether the gene expression profiles of normal and neoplastic organoid samples from the same patient are similar due to the proliferation-inducing cell culture conditions that may mask the differences between neoplastic and non-neoplastic cells. To answer this question, we analyzed the differential gene expression between organoids derived from CMTs and those derived from normal mammary tissues. As a result, we found that 109 genes are significantly (*p*-value < 0.05 and |LFC| > 1.0) upregulated (labeled in red in Figure 2c) and 129 are significantly downregulated (labeled in green in Figure 2c). After GO enrichment analysis, we observed that, on the one hand, many genes significantly upregulated in organoids derived from CMTs are involved in growth factor binding, signaling receptor binding, and peptidase regulator activity (Figure 2d). On the other hand, we found that genes significantly downregulated in organoids derived from CMTs are mainly involved in transmembrane receptor protein tyrosine kinase activity and multicellular organism development (Figure 2d). Then, we investigated whether the candidates from the CRISPR screens were differentially expressed in organoids derived from CMTs compared with organoid samples derived from normal mammary tissues. Most of the candidates were expressed in the organoid lines but, in all cases, organoid line-specific essentiality (genes that are functionally essential for ORG-63-C but not for ORG-63-N) did not reflect differential expression (see Appendix A).

In summary, despite substantial differences between organoids and primary tissues regarding gene expression, our data show that the candidates from the CRISPR/Cas9 screens are also expressed in the primary tumors and therefore are not just induced by the in vitro culture conditions. Moreover, our results suggest that increased essentiality is not represented by increased gene expression levels in tumor tissues or tumor-derived organoids. In addition, when comparing the results of our two independent screens, we identified CDK2 as a common hit for both screens and set out to validate this candidate (Figure 2).
Figure 2Candidates from the CRISPR/Cas9 screens are expressed in primary tumor tissues and PDOs but very rarely differentially expressed compared to the non-neoplastic mammary tissues.
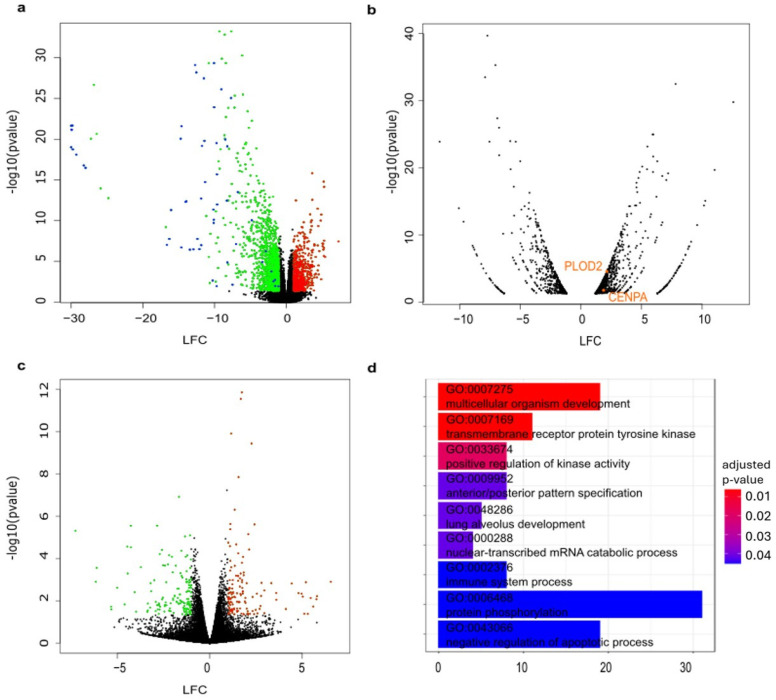

Volcano plot representing downregulated (log fold change = LFC < 0) and upregulated (LFC 0) genes between all CMT samples compared with all normal mammary tissue samples. Significantly (*p*-value < 0.05 and |LFC|> = 1.0) downregulated (green) and upregulated (red) are color coded. Keratins and keratin-associated proteins are labeled in blue.Volcano plot representing significantly downregulated and upregulated (*p*-value < 0.05 and |LFC| > 1.0) genes between D63_3_CMT and D63_nor. Candidates from the CRISPR/Cas9 screens that are differentially expressed between D63_3_CMT and D63_nor are labeled in orange.Volcano plot representing downregulated (LFC < 0) and upregulated (LFC > 0) genes between three organoid lines derived from carcinomas compared with their matched organoids derived from normal mammary tissues. Significantly (*p*-value < 0.05 and |LFC| > 1.0) downregulated (green) and upregulated (red) are color coded.Bar plot showing Gene Ontology (biological processes) analysis results of all differentially expressed genes between three organoid lines derived from carcinomas compared with their matched organoids derived from normal mammary tissues. The color of each bar represents the *p*-value of each term involved in the analysis, and the bar size represents the gene counts for this term.

### 3.5. CDK2 May Be a Druggable Vulnerability for CMTs

To overcome the limitation of gene essentiality, where a direct genetic knock-out unequivocally leads to cell death, we proceeded to validate our candidate with known inhibitors. Unfortunately, a sole inhibitor for CDK2 does not exist [45], as many CDK2 inhibitors also have an affinity for CDK1, which is highly essential for normal tissues [46]. PF3600 is a recently developed selective CDK2/4/6 inhibitor [47]. As *CDK4* and *CDK6* were also targeted by our CP1736 “druggable” library, we first checked whether they appeared to be essential in ORG-63-N. This was not the case with an LFC > −1 or *p*-value > 0.05 for ORG-63-N (Table 3), and we proceeded to test this inhibitor on our two organoid lines.

A drug viability assay showed that ORG-63-C was more sensitive to PF3600 than ORG-63-N, validating *CDK2* as a vulnerability of ORG-63-C that can be targeted (Figure 3a). In addition, the IC50 found (0.53 µM and 0.16 µM) were compatible with drug concentrations tolerated in mice [48]. Although ORG-63-C exhibited approximately a 3-fold increased sensitivity, the difference in IC50 is modest, suggesting that more specific inhibitors may provide clearer specificities in the future.

In addition, we tested three other hits from the druggable screen—TOP2B, TTK, and HSP90AA1—using the commercially available inhibitors doxorubicin [49], reversine [50], and luminespib [51], respectively. ORG-63-C appeared more sensitive to doxorubicin than ORG-63-N (Figure 3b), while no difference was observed for reversine (Figure 3c) and luminespib (Figure 3d).

While doxorubicin remains toxic for ORG-63-N (Figure 3a) as it also targets the shared essential topoisomerase TOP2A, its lethal effect appears stronger for ORG-63-C, for which TOP2B is essential in addition. This could indicate the presence of compensatory mechanisms in ORG-63-N, which may limit the drug’s efficacy in this line.

Conversely, ORG-63-C did not appear more sensitive to reversine (Figure 3c), chosen initially as a TTK inhibitor. However, reversine is also an inhibitor of Aurora A and B kinases. According to our druggable screen results (Appendix A), AURKB and AURKA are essential (LFC < −1 and *p*-value < 0.05) for both organoid lines, which can explain the similar response to reversine when treating ORG-63-N and ORG-63-C. We also did not see any difference with luminespib (Figure 3d), an inhibitor of subunits α and β of HSP90 [52]. Known off-target effects of luminespib on different kinases, which might be essential for both organoid lines, could explain this result [53].

Overall, after identifying CDK2 as a common hit in both unbiased screening approaches, our drug assay using PF3600 validated CDK2 function as a potentially interesting essentiality for CMTs to be targeted with a selective inhibitor (Figure 3).
Figure 3ORG-63-C is more sensitive to PF3600 and doxorubicin than ORG-63-N.
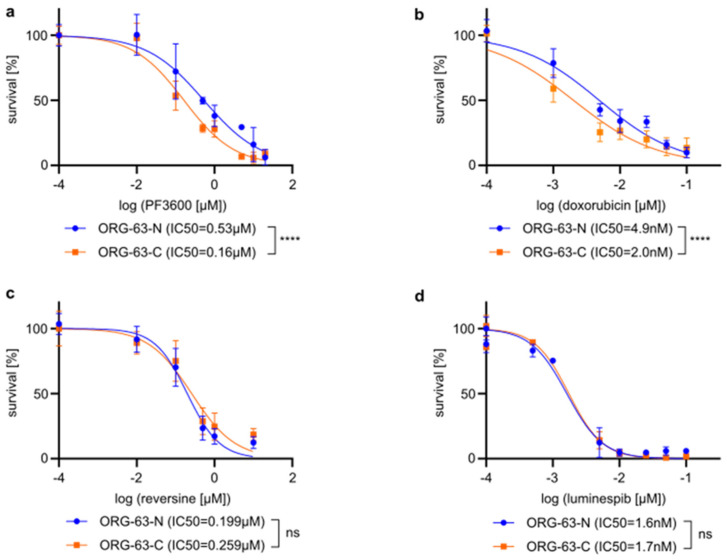

Dose-response curves illustrating the effect of PF3600 on cell viability in organoid lines ORG-63-C and ORG-63-N. Error bars represent the standard deviation (SD) of three independent experiments. Statistical analysis of the log-transformed IC50 values was performed using an unpaired *t*-test, **** *p* < 0.0001. IC50s are indicated in the figure legend.Dose-response curves illustrating the effect of doxorubicin on cell viability in organoid lines ORG-63-C and ORG-63-N. Error bars represent the standard deviation (SD) of three independent experiments. Statistical analysis of the log-transformed IC50 values was performed using an unpaired *t*-test, **** *p* < 0.0001. IC50s are indicated in the figure legend.Dose-response curves depicting the effect of reversine on cell viability in organoid lines ORG-63-C and ORG-63-N. Error bars represent the standard deviation (SD) of three independent experiments. Statistical analysis of the log-transformed IC50 values was performed using an unpaired *t*-test, ns = non-significant. IC50s are indicated in the figure legend.Dose-response curves depicting the effect of luminespib on cell viability in organoid lines ORG-63-C and ORG-63-N. Error bars represent the standard deviation (SD) of three independent experiments. Statistical analysis of the log-transformed IC50 values was performed using an unpaired *t*-test, ns = non-statistical. IC50s are indicated in the figure legend.

## 4. Discussion

In this study, conducted with CRISPR/Cas9 screenings using two custom-designed canine sub-libraries on paired canine mammary PDOs derived from both the non-neoplastic and cancerous mammary tissues of a single donor. While CRISPR/Cas9-based approaches have been used in carcinogenesis studies with colon organoids [25,54,55], to the best of our knowledge, the use of paired non-neoplastic and cancerous mammary PDOs was first demonstrated in our previous study, due to the challenges in obtaining normal human mammary tissue from HBC patients [11] Our approach helps mitigate the bias associated with inter-individual genetic variability.

Following this unbiased approach, CDK2 was identified as a common candidate in both screens, essential for the malignant PDOs but dispensable for the normal PDOs. This context-specific essential gene may be a useful therapeutic target with high effectiveness and minimal side effects (concept of synthetic lethality [18].

CDKs 1, 2, 4, and 6 play an essential role in cell cycle regulation [56] and control progression through each of the four phases of the cell cycle (G1, S, G2, and M). While targeting the cell cycle as an opportunity for cancer treatment is not novel in human oncology, it remains underexplored in a veterinary setting [57]. For example, selective pharmacological inhibition of CDK4/6 showed promising results in HBC, with the CDK4/6 dual inhibitor palbociclib being approved in combination with the aromatase inhibitor letrozole in 2015 for treating ER+ HBC patients [58]. However, patients often develop resistance to selective CDK4/6 inhibitors with different mechanisms involved, among which is the activation of MYC and CDK2. Combined inhibition of CDK2/4/6 shows promising in vitro and in vivo results [48], and the CDK2/4/6 inhibitor PF3600 is currently being tested in a phase I clinical trial for different types of HBC [59]. These insights suggest that further exploration of CDK2 inhibition may be promising in CMT therapy.

In parallel with CDK inhibition, CRISPR/Cas9 holds potential for directly targeting specific genes, offering a strategy for overcoming the resistance. However, the use of CRISPR/Cas9 for in vivo treatments is still in its infancy and remains limited due to concerns over safety, off target effects, and delivery efficacy [60]. Nonetheless, its application in preclinical models, such as PDOs, offers valuable insights on how gene editing tools could be leveraged for potential therapeutic interventions in the future.

Conventional chemotherapy does not appear to provide a clear advantage for dogs suffering from CMTs [5], and owners are highly motivated to find novel options for the treatment of cancer in their pets. Therefore, preclinical studies using CMT organoids could help predict the treatment efficacy before clinical trials. For example, known inhibitors, such as PF3600, can be tested first in CMT organoids to predict therapy efficacy, followed by testing in CMT-bearing dogs to assess the safety and efficacy, leading to translatable results [61]. Furthermore, as dogs share the same environment as their owners and are exposed to the same carcinogens as humans, the tumors they develop might be influenced by similar epigenetic regulation processes.

Studies investigating the epigenetic landscape in canine cancer are scarce, very few focus on CMTs and have enough samples to draw solid conclusions (for a review, see [62]). For example, a study found only one CMT sample (out of fifteen) displaying BRCA1 hypermethylation, which might explain the downregulation of BRCA1 in a few cases [63,64]. No variation in methylation patterns was found between ERα-positive CMTs and ERα-negative CMTs, indicating that deregulation of ERα in CMTs, contrary to HBC, is not attributed to *ERα* promoter methylation [65]. Genome-wide methylation studies have been performed, but only for two dogs [66]. EZH2 (enhancer of zeste homolog 2), a crucial epigenetic regulator overexpressed in HBC, was overexpressed in CMTs, and its expression level (determined with immunohistochemistry) was associated with the degree of malignancy of the CMTs [67].

It has also been suggested that complex carcinomas originate from epigenetic rather than genomic alterations. Indeed, a pioneering study showed that complex carcinomas (four cases), compared to simple carcinomas (seven cases), displayed several epigenetic dysregulations, such as downregulation of chromatin-modification genes or enriched activating histone modification H4-acetylation, while showing a reduction in the repressive histone modification H3K9me3 [68]. The first evidence of microRNA (miRNA) dysregulation in canine tumors was described for CMT samples, where both *miR-29* and *miR-29b* were upregulated in CMTs [69]. Ten miRNA (*cfa-let-7c*, *cfa-miR-10b*, *cfa-miR-26a*, *cfa-miR-26b*, *cfa-miR-29c*, *cfa-miR-30a*, *cfamiR-30b*, *cfa-miR-30c*, *cfa-miR-148a*, and *cfa-miR-299*) showed significant different expression in metastatic and non-metastatic CMTs [70]. CMT cells also appear to shed exosomes containing differentially expressed miRNA compared to normal cells [71]. These few descriptive studies already show interesting aspects of epigenetic regulation in CMTs and highlight the urgent need for further studies investigating the epigenetic players in CMTs. Our epigenome screen yielded exciting new candidates, and validation of those hits will bring our understanding of CMT tumorigenesis forward.

In the future, however, for an optimal validation of our different candidates, we need to overcome the limitation of gene essentiality, where a direct genetic knock-out unequivocally leads to cell death. To avoid this problem, in this study, we worked with known inhibitors when they were available. However, they are not always specific and target either other genes (off-target effects) or both subunits of the same protein. In this context, a gene-editing approach to confirm the phenotype induced by the studied mutation is ideal. For this purpose, knockdown systems are helpful for studying essential genes. RNA interference with short-hairpin RNAs (shRNAs) or short-interfering RNAs (siRNAs) can be used in organoids, even if the process is more challenging than in 2D cell cultures [72,73]. However, they show lower efficacy and are also prone to off-target effects. Different inducible systems to trigger gene knockouts or knockdowns have been explored, some of which remain to be adapted to organoid cultures [74]. For example, an inducible gene knockdown method with a doxycycline-inducible system was established in a human lung PDO [75]. Another powerful method that could be applied to organoid cultures is the CRISPR-FLIP, which is based on the homology-directed repair mediated insertion of a Cre/loxP invertible cassette into an intronic region of a target gene [76]. Then, the auxin-induced degron (AID) approach showed that AID tagging, in combination with CRISPR/Cas9 mediated gene targeting, results in the rapid degradation of proteins [77,78]. However, this technique remains to be established in organoid culture. Therefore, many different approaches can be tested for the functional analysis of essential genes in organoids.

Our RNA sequencing analysis revealed significant differences between organoids and primary tissues when focusing on gene expression. This poses a challenge in fully mimicking the primary tissue in vitro. A direct transcriptomic comparison to paired primary tissue has not yet been performed for HBC-derived organoids, with RNA sequencing data from organoids only being compared to published datasets [79]. Despite the advances in organoid technology, and their closer resemblance to primary tissues compared to 2D cell cultures, there are still some limitations when focusing on gene expression.

These differences can be explained through different aspects. First of all, the growth medium used for the organoid culture enhances the growth of epithelial cells, and the other non-epithelial components of the primary tumor are lost. Therefore, gene expression from stromal cells of the tumor microenvironment that do not grow in the traditional organoid cultures may contribute to the differences in gene expression between primary tumors and organoids and possibly lead to an unclear distinction between normal and neoplastic cells. In the future, spatial transcriptomics could be useful in resolving this issue [80]. Indeed, this next-generation molecular profiling technique allows the mapping of the whole transcriptome with morphological context on the slide in FFPE or fresh frozen tissues. Then, the primary samples’ heterogeneity might account for the transcriptomic differences between organoids and primary tissues. Indeed, different tissue pieces are used for RNA sequencing and organoid derivation.

In addition, it is sometimes impossible to macroscopically differentiate between non-neoplastic mammary tissues and pre-neoplastic or benign lesions when collecting tissues for the organoid biobank. As CMT tumorigenesis is thought to follow a stepwise evolution from adenomas to carcinomas [6], comparing the gene expression levels of adenomas with non-neoplastic mammary tissue samples might shed light on this process. Next, messenger RNA (mRNA) expression might vary depending on the tumor stage or the samples’ freezing conditions (some CMT samples were shipped overnight and kept at 4 °C during transport, leading to possible mRNA degradation).

Despite the differences in gene expression between organoids and primary tissues, the gene candidates from the CRISPR/Cas9 screens are also expressed in the primary tumors, suggesting that these differences are not solely due to the in vitro culture conditions. However, further validation would be necessary to confirm this to rule out potential artifacts from the culture systems. Additionally, our results suggest that increased essentiality is not represented by increased gene expression levels in tumor tissues or tumor-derived organoids. Indeed, standard gene expression profiling and other “omics” techniques do not usually pick up on subtle modifications. If some essential genes can be overexpressed, as is sometimes the case [81], it is not necessary because gene function does not automatically reflect itself on gene expression. If we had performed a genome-wide CRISPR/Cas9 screen, we would probably have found more essential genes that were also upregulated, but with such a small dataset of essential genes from our pooled libraries, it is unsurprising that very few of them are overexpressed. Similarly, Hu et al. [82], focusing on colon adenocarcinoma published datasets, found that, among 1166 essential genes identified with genome-scale CRISPR/Cas9 screening data, only 261 were significantly upregulated, based on RNA sequencing data from TCGA [83] and GTEx [84]. This suggests that combining RNA sequencing with functional screening approaches is critical for an understanding of tumor biology.

In summary, our data show the first analysis of two drop-out screens conducted with custom CRISPR/Cas9 libraries for canine PDOs developed from spontaneous CMTs and its matched healthy mammary tissue. Our approach has validated CDK2 as a therapeutic vulnerability for CMTs. Moreover, we showed the power of functionally testing gene essentiality in matched neoplastic versus non-neoplastic PDOs of an individual patient. This allows functional testing of hundreds of genes that can be targeted and provides valuable information about potential therapeutic approaches for a particular patient.

## Figures and Tables

**Table 2 vetsci-12-00183-t002:** Hits essential in ORG-63-C but dispensable in ORG-63-N (druggable screen).

Gene	Gene Description	*p*-Value_N	lfc_N	*p*-Value_C	lfc_C
HSP90AA1	Heat shock protein HSP 90-alpha	0.70337	0.18138	0.02346	−1.547
FANCD2	Fanconi anemia group D2 protein	0.603	−0.22887	0.04947	−1.1426
CDK2	Cyclin-dependent kinase 2	0.53595	−0.27009	0.039806	−1.4028
TOP2B	DNA topoisomerase 2-beta	0.39481	−0.0917	0.0087646	−1.5298
RPTOR	Regulatory-associated protein of mTOR	0.39167	−0.45626	0.048055	−1.0535
BCL2L1	Bcl-2-like protein 1	0.31849	−0.26447	0.013289	−1.2187
MYBL1	Myb proto-oncogene like 1	0.31416	−0.5715	0.024879	−1.6852
TTK	Serine/threonine-protein kinase ttk/mps1	0.28795	−0.61574	0.013661	−1.584
FANCC	Fanconi anemia group C protein	0.24945	−0.54887	0.0073124	−1.9841
POLE	DNA polymerase ε catalytic subunit A	0.24504	−0.27922	0.013083	−1.3574
CDK12	Cyclin-dependent kinase 12	0.23879	−0.16167	0.008662	−1.9851
GMPS	GMP synthase [glutamine-hydrolyzing]	0.23684	−0.25904	0.035056	−1.4074
CCND3	G1/S-specific cyclin-D3	0.18021	−0.86676	0.017937	−1.3585
USP5	Ubiquitin carboxyl-terminal hydrolase 5	0.16569	−0.87191	0.023271	−1.2658
NFKBIA	NF-kappa-B inhibitor alpha	0.14329	−0.99465	0.0078846	−2.0144
PI4KB	Phosphatidylinositol 4-kinase beta	0.10617	−0.86682	0.036475	−1.4132
FANCE	Fanconi anemia group E protein	0.09315	−0.67103	0.026213	−1.027

**Table 3 vetsci-12-00183-t003:** MAGeCK screen results for CDK2, 4 and 6.

Library	Gene	*p*-Value_N	lfc_N	*p*-Value_C	lfc_C
CP1736	CDK2	0.53595	−0.27009	0.039806	−1.4028
	CDK4	0.92691	0.28251	0.16707	−0.84773
	CDK6	0.10149	−1.1315	0.0024321	−1.6764
CP1737	CDK2	0.31995	−0.74951	0.024024	−1.3541

## Data Availability

Both custom CRISPR/Cas9 sub-libraries used in this publication are available in the Appendix A. Both raw and processed RNA sequencing data have been deposited in the European Nucleotide Archive (ENA) database under accession code PRJEB85459. Additional data can be made available upon reasonable request to the corresponding author.

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
