# Peer review of "Individualized Pooled CRISPR/Cas9 Screenings Identify CDK2 as a Druggable Vulnerability in a Canine Mammary Carcinoma Patient"

_vetsci, 2025, doi:10.3390/vetsci12020183_

Round 1

Reviewer 1 Report

Comments and Suggestions for Authors

The manuscript is an interesting study about the use of patient derived organoids (PDOs) by performed the CRISPR/Cas9 screenings to identify susceptible genes in canine mammary carcinomas. For this, were studied cells both mammary tumours and normal mammary tissue from the same dog, which led to determine that the gene CDK2 as vulnerable in the tumour’s cells. In general, the manuscript describing the methodology used in detail, discussion and conclusions are convincing and considering the sequence of previously published works by the authors on canine mammary tumours organoids, there are some suggestions to improve the manuscript:

-Introduction: in lines 60, 63,66, 87,88, 95 try to avoid repetitive self-citations. You can use your previous studies to some specific relevant results but avoid being repetitive.

-Histopathology:  The histological classification of tumours is mentioned but is not reflected in any table, not even in the supplementary ones, if it is possible to mention the histological types observed.

-Why do you use the dog 63, and what was the histological type of tumour analysed? It is important to refer it in the methodology.

-Lines 303, 307 and 318, please correct the references according to the guidelines.

-Tables: please structure the tables according to the guidelines of the journal.

-Supplementary information: some information is missed for example tables S3, S5, S7, S8 and S9.

Reviewer 2 Report

Comments and Suggestions for Authors

In this study, the authors attempt to identify mammary tumor-specific therapeutic targets by obtaining tumor PDO and normal mammary PDO from the same individuals affected by mammary tumors and comparing their sensitivity to therapeutic agents. This is an interesting and challenging study and has the potential to contribute to clinical applications. I have some comments which I believe improve the significance and readability of the paper.

1.     The authors should provide the reader with information regarding the details of what canines the materials used in this study were obtained from. Indeed, it is known that tumor incidence is affected by canine strain and age, and it is likely that these factors will also influence this study. It should also be noted what kind of medical history and treatment history this canine has.

2.     The authors identified CDK2 as a treatable vulnerability in this study. The authors often mention other candidates besides CDK2 in the paper, but it is difficult to understand why only CDK2 was examined in detail in this study, and it would be desirable to state this reason in a clear and concise manner.

3.     The authors identified CDK2 as a treatable vulnerability in this study. The authors often mention other candidates besides CDK2 in the paper, but it is difficult to understand why only CDK2 was treated in this study, and it would be desirable to state this reason in a clear manner. Also, if it is possible to identify promising candidates to be considered in the future, they should be described in more detail.

4.     The authors state that this is the first report using this technique in canine mammary tumors, but if there have been reports of similar concepts in other canine tumors or in human breast cancer, why not compare this study with them? This would make the novelty and uniqueness of this study clearer.

5.     Is there any possibility that CRISPR/Cas9 itself, which was used for screening in this study, could be used for treatment?

6.     The following statement in the introduction is an oversimplification. In reference 7, isn't it stated that the results are about the cohort examined in the study? Shouldn't this be explained more carefully? For example, the effect of doxorubicin is seen in Figure 3b of this study. Could this drug be involved in DNA damage or homologous recombination?

Moreover, characteristic HBC mutational signatures, such as those resulting from defects in APOBEC cytidine deaminases or homologous recombination repair deficiency [12], do not play a significant role in CMT carcinogenesis [7].

Reviewer 3 Report

Comments and Suggestions for Authors

The manuscript is very well written and presents promising results. The topic is relevant to the veterinary literature and explores a very important condition of dogs.

A few considerations and questions are listed below:

Abstract

Consider rewriting the latter paragraph in order to make it more concise and focus only on the observed results.

Keywords

Consider avoiding repetition of terms between keywords and title. Repeated keywords such as ‘CRISPR/Cas9 screening’ and ‘canine mammary tumor’ could be changed for others not previously used which could also improve TIAB searches (e.g., ‘gene-editing tools’, ‘mammary tumors’, etc.).

INTRODUCTION

Avoid overuse of the term ‘moreover’.

Page 2:

Line 51: Despite correct, consider using more recent references - especially those based on cancer registries.

Lines 80-81: Avoid use of absolute terms/sentences. Consider rewriting the sentence (e.g., … ‘treatment options are frequently/often unsuccessfull’).

MATERIALS AND METHODS

Lines 139-140: Since there are divergences among groups regarding studies in mammary tumors and no higher mediation such as the World Health Organization/World Organization of Animal Health validating the data, no gold standards can be truly indicated. Thus, consider rewriting the sentence (e.g., …’classified and graded as previously described’).

RESULTS

A more wide description of the pathologic evaluation is lacking. Since mammary tumors in dogs can be highly heterogeneous and due to the various histotypes - with different behaviors and characteristics - it is very important to describe such features in order to better understand the vulnerabilities and changes, especially because this manuscript addresses organoids from a single patient without other controls. 

DISCUSSION

Clearly identified speculation can always add to the value of a scientific report. However, consider avoiding the use of absolute terms/sentences. Also, consider rewriting the sentence (e.g., ‘We performed two distinct pooled…).

Lines 657-690 and 712-738: The paragraphs are considerably long. Consider rewriting in a more concise way or split in two different paragraphs (particularly in lines 681 and 731) in order to make it better for readers.

Lines 747-749 and 758-759: Clearly identified speculation can always add to the value of a scientific report. However, care should be taken about overspeculation.

General considerations:

The manuscript is well written. Also, the study shows potential but needs to be reviewed for further considerations – particularly the discussion section and the pathologic evaluation data. More recent references (including those from cancer registries) could be included. Additionally, it would be better to address the evaluation of a single patient in the title in order to  make it more clear to readers (e.g., …’druggable vulnerability in a canine mammary carcinoma patient’).
